# Safety and Effectiveness of Intravenous Iron Therapy in Patients Supported by Durable Left Ventricular Assist Devices

**DOI:** 10.3390/jcm11133900

**Published:** 2022-07-04

**Authors:** Carli J. Peters, Thomas C. Hanff, Michael V. Genuardi, Robert Zhang, Christopher Domenico, Pavan Atluri, Jeremy A. Mazurek, Kim Urgo, Joyce Wald, Monique S. Tanna, Supriya Shore, Michael A. Acker, Lee R. Goldberg, Kenneth B. Margulies, Edo Y. Birati

**Affiliations:** 1Department of Medicine, Perelman School of Medicine, University of Pennsylvania, Philadelphia, PA 19104, USA; carli.peters@pennmedicine.upenn.edu (C.J.P.); thomas.c.hanff@gmail.com (T.C.H.); michael.genuardi@pennmedicine.upenn.edu (M.V.G.); robert.zhang@nyulangone.org (R.Z.); jeremy.mazurek@pennmedicine.upenn.edu (J.A.M.); joyce.wald@pennmedicine.upenn.edu (J.W.); monique.tanna@pennmedicine.upenn.edu (M.S.T.); lee.goldberg@pennmedicine.upenn.edu (L.R.G.); kenneth.margulies@pennmedicine.upenn.edu (K.B.M.); 2Division of Cardiology, Perelman School of Medicine, University of Pennsylvania, Philadelphia, PA 19104, USA; christopherdomenico@pennmedicine.upenn.edu; 3Department of Biostatistics, Epidemiology and Informatics, Perelman School of Medicine, University of Pennsylvania, Philadelphia, PA 19104, USA; 4Cardiovascular Outcomes, Quality, and Evaluative Research Center, University of Pennsylvania, Philadelphia, PA 19104, USA; 5Department of Pharmacy, Hospital of The University of Pennsylvania, Philadelphia, PA 19104, USA; 6Department of Cardiothoracic Surgery, Perelman School of Medicine, University of Pennsylvania, Philadelphia, PA 19104, USA; pavan.atluri@pennmedicine.upenn.edu (P.A.); kim.urgo@pennmedicine.upenn.edu (K.U.); michael.acker@pennmedicine.upenn.edu (M.A.A.); 7Cardiovascular Division, University of Michigan, Ann Arbor, MI 48109, USA; shores@med.umich.edu; 8The Lydia and Carol Kittner, Lea and Banjamin Davidai Division of Cardiovascular Medicine and Surgery, Padeh-Poriya Medical Center, Azrieli Faculty of Medicine, Bar-Ilan University, Ramat Gan 5290002, Israel

**Keywords:** intravenous iron, left ventricular assist devices, iron deficiency

## Abstract

Aims: While it is common practice to use intravenous (IV) iron in patients with left ventricular assist devices (LVADs) and iron deficiency, there is insufficient evidence regarding outcomes in this patient population. We evaluated the safety and effectiveness of IV iron therapy in patients supported by LVADs with iron deficiency. Methods: We performed a retrospective analysis of iron deficient patients on continuous LVAD support at a large academic center between 2008 and 2019. Patients were divided into two cohorts based on IV iron sucrose treatment. The primary endpoint was hemoglobin at 12 weeks. Secondary endpoints were mean corpuscular volume (MCV) and New York Heart Association (NYHA) class at 12 weeks. Safety endpoints included hospitalization, infection, pump thrombosis, arrhythmia, and gastrointestinal bleed. Models were weighted by the inverse probability of receiving IV iron using a propensity score, and endpoints were adjusted for their corresponding baseline values. Results: Among 213 patients, 70 patients received IV iron and 143 patients did not. Hemoglobin at 12 weeks was significantly greater among those treated (intergroup difference: 0.6 g/dL; 95% CI, 0.1 to 1.1; *p* = 0.01), while MCV was similar in both groups (intergroup difference: 0.7 μm^3^; 95% CI, −1.3 to 2.7; *p* = 0.50). NYHA class distribution at 12 weeks was significantly different (odds ratio for improvement: 2.84; 95% CI, 1.42 to 4.68; *p* = 0.003). The hazards of adverse events in each group were similar. Conclusions: In patients with LVADs and iron deficiency, treatment with IV iron sucrose was safe and associated with improvements in functional status and hemoglobin.

## 1. Introduction

Durable mechanical assist devices have emerged as an important therapeutic option for patients with advanced heart failure, both as a bridge to heart transplantation and as a destination therapy. Despite significant recent improvements in survival and device durability with continuous flow left ventricular assist devices (LVADs), a high burden of morbidities and frequent hospitalizations remain a significant hardship for patients [1,2,3]. Moreover, approximately 20% of the patients on LVAD support continue to have limited functional status, with a New York Heart Association (NYHA) functional class III or IV [4], despite cardiac rehabilitation and efforts to optimize pump function [5,6].

Iron deficiency is present in up to 50% of patients with advanced heart failure [7] and is associated with increased heart failure admissions, reduced exercise tolerance, and reduced quality of life [8,9,10]. Treatment with intravenous (IV) iron in patients with heart failure and iron deficiency, with and without anemia, has been associated with improved functional status and quality of life [11,12]. Based on this evidence, current American Heart Association/American College of Cardiology and European Society of Cardiology guidelines recommend IV iron therapy for heart failure patients with NYHA II and III with evidence of iron deficiency [13,14].

Iron deficiency anemia remains prevalent among patients with advanced heart failure supported by durable LVAD and is associated with increased risk of hospitalization and mortality [15]. As a consequence, many centers have extended the recommended use of IV iron in heart failure to patients with an LVAD who have iron deficiency. However, IV iron therapy may have unique risks in LVAD patients, and there is insufficient evidence regarding the benefit and safety of IV iron therapy in this patient population.

In this retrospective, inverse probability weighted analysis, we evaluated the safety and effectiveness of IV iron therapy in LVAD patients with iron deficiency.

## 2. Methods

### 2.1. Study Design

We performed a retrospective study among patients supported with a continuous flow LVAD and concomitant iron deficiency. Data were collected from electronic medical records of patients treated at the Hospital of the University of Pennsylvania between 2008 and 2019. Both ambulatory and admitted patients with an LVAD (HeartMate II (Abbott, Abbott Park, IL, USA), HeartMate III (Abbott, Abbott Park, IL, USA), and HeartWare (Medtronic Inc., Minneapolis, MN, USA)) and iron deficiency were included. Iron deficiency was defined according to the current literature as either (1) ferritin < 100 ug/L or (2) ferritin 100–299 ug/L with transferrin saturation <20% [11,12]. We focused on iron deficiency beginning at 90 days after LVAD implantation to avoid peri-operative events that may significantly confound the relationship between IV iron and endpoints.

Patients who met these criteria were divided into two cohorts based on IV iron treatment, which is formulated as IV iron sucrose at our institution. The treatment cohort consisted of patients who were treated with IV iron, while the control cohort consisted of patients who were not treated with IV iron. Treatment was defined per clinical protocol with dosing administrations of 200 mg for 5 doses or 300 mg for 3 doses. Inpatient treatment was dosed daily, while outpatient treatment was dosed weekly. Patients who did not receive a complete dose were placed in the treatment cohort as intention-to-treat. If patients were treated multiple times with IV iron sucrose at the institution, only their first treatment was included. Among this cohort of patients eligible for IV iron, it is difficult to ascertain why certain patients were treated and others were not. Potential factors leading to an increased likelihood of treatment may include profound iron deficiency and anemia; patients with lower ferritin, lower hemoglobin, and lower mean corpuscular volume (MCV) may have been prompted the physician to treat the patient with IV iron. In a similar fashion, patients with known history of gastrointestinal bleeds may have precipitated treatment.

Baseline demographics and characteristics were obtained via review of electronic medical records for each patient. Baseline data that was available closest but prior to the date of the iron deficiency diagnosis were collected. This study was approved by the University of Pennsylvania Institutional Review Board. The requirement for specific informed consent for this study was waived on the basis of minimal privacy risk. De-identified data and analytical methods that support the findings of this study are available from the corresponding author upon request.

### 2.2. Effectiveness and Safety Endpoints

The primary endpoint was the plasma hemoglobin concentration at 12 weeks following IV iron therapy. Secondary endpoints were the plasma MCV and the NYHA class at 12 weeks. Endpoints were recorded closest to 12 weeks (±4 weeks) after diagnosis of iron deficiency for the control group or treatment with IV iron for the treatment group. This timeframe was chosen as previous studies have demonstrated significant endpoint improvements within 12 weeks of treatment initiation [11,12], and our patient population has frequent follow-up visits at 3-month intervals, allowing for follow-up data to be readily available. Every patient in the cohort had a plasma hemoglobin concentration, plasma MCV, and NYHA class within the specified timeframe. The NYHA class is a part of the template assessment outlined by the advanced heart failure attending physician. Ferritin and iron saturation were not included as endpoints because these measurements are not routinely repeated after IV iron treatment.

Safety endpoints encompassed some well-known, postmarketing adverse events for IV iron including the rate of anaphylaxis, ventricular and atrial arrhythmias, infection, gastrointestinal bleeding and ventricular assist device (VAD) pump thrombosis within 90 days of the iron deficiency diagnosis. Infection was defined by treatment with antimicrobials. Clinically significant bleeds were defined as requiring transfusion of red blood cells or admission with bleeding as the principal reason. We also assessed the rate of hospitalization and cardiovascular admissions within 90 days of diagnosis. Cardiovascular admissions were identified as admissions with an ‘I’-group ICD-10 code or the corresponding ICD-9 code.

### 2.3. Statistical Analysis

Baseline characteristics for categorical data were described using number and percentage and compared using Pearson’s chi-squared, while continuous variables were expressed as medians and interquartile ranges and compared using the Wilcoxon rank-sum. For continuous endpoints, comparisons between the two groups were performed using linear regression. For ordinal endpoints, differences in distribution between the two study groups were tested by means of ordinal logistic regression. Time-to-event analyses were performed using Kaplan–Meier estimators and log-rank tests. Hazard ratios (HRs) and corresponding 95% confidence intervals (CI) were obtained from Cox proportional hazards models. Proportional hazard assumptions were evaluated using Schoenfeld residuals. A two-side Type I error rate of 0.05 was considered significant. All statistical analyses were performed using the Stata MP 16.0 software (College Station, TX, USA).

Three modeling strategies were used to estimate the direct effect of IV iron on each dependent outcome variable. First, to control for differences in baseline patient characteristics that may confound estimates of the association between IV iron and each outcome, regression and hazard models were weighted by the inverse probability of the propensity score. The propensity score represents the probability of receiving IV iron treatment conditional on baseline covariates in a probit model [16]. Covariates in the propensity score model were defined prior to the analysis and included age, sex, race, body mass index (BMI), inpatient vs. outpatient status, LVAD model, NYHA class, heart failure etiology, days between LVAD implantation and diagnosis, baseline labs including ferritin, hemoglobin, MCV, creatinine, aspartate aminotransferase (AST), and alanine aminotransferase (ALT), binary history of gastrointestinal bleed, diabetes mellitus, and chronic obstructive pulmonary disease (COPD), and binary use of baseline medications including oral iron, anticoagulant, antiplatelet, beta-blocker, angiotensin-converting enzyme inhibitors (ACE) or angiotensin II receptor blockers (ARB), digoxin, proton pump inhibitor, and H2 blocker. Covariate imbalance before and after matching was assessed for all baseline covariates by estimating standardized differences, which are the average differences between means in standard deviation units [16]. Standardized differences less than 25.0% for a given covariate indicate small imbalances [16]. Second, the model for each dependent 12-week outcome variable included adjustment for that outcome’s corresponding value at baseline. This strategy mitigates potential regression to the mean and requires the least number of statistical assumptions when considering the effect of baseline values on 12-week outcome [17]. Lastly, for each outcome, we were primarily interested in estimating the direct causal effect of IV iron treatment on the outcome. However, the receipt of IV iron at baseline may change the likelihood of later blood transfusions or the number of units of transfusion given, and blood transfusions could impact each of the dependent outcome variables. Thus, blood transfusions may serve as a mediator of the effect of IV iron on outcome. Therefore, in order to estimate the direct effect of IV iron on outcome, we also adjusted for the total number of packed red blood cell (PRBC) units administered between baseline and week 12. Statistically, this eliminates the indirect effect of IV iron on outcome mediated by blood transfusions.

## 3. Results

### 3.1. Study Population

We identified 528 patients supported by LVAD at our institution between 2008 and 2019, of whom 213 (40%) were iron deficient. Of these patients, 70 patients were treated with IV iron and 143 patients were not. There were 3 patients in the treatment group and 5 patients in the control group with incomplete data due to loss of follow-up. As this was less than 5% of the total study population, only complete cases were included in the propensity score analysis. Ultimately, 67 treatment patients and 138 control patients were analyzed (Figure 1).

Prior to inverse probability weighting, there were significant differences between the two groups in several of the baseline variables (Table 1). Notably, the median baseline hemoglobin was lower in the treatment group 9.2 g/dL [IQR 8.5, 10.6] than the control group 11.0 g/dL [9.6, 12.8] (*p* < 0.001). Likewise, the median baseline MCV was lower in the treatment group 79.0 μm^3^ [73.0, 86.0] than the control group 86.0 μm^3^ [81.0, 90.0] (*p* < 0.001). The distribution of NYHA class baseline was significantly different (*p* = 0.006) with 49% of the treatment group and 26% of the control group in class III or IV. The number of patients who were inpatient at time of diagnosis was significantly higher in the treatment group 69% vs. 31% (*p* = <0.001). All of the baseline characteristics listed in Table 1 were used in the propensity score and, after weighting, resulted in the cohorts having no significant differences between them with *p*-values >0.05.

### 3.2. Primary and Secondary Endpoints

At 12 weeks, the mean hemoglobin was 11.0 g/dL (95% CI, 10.5 to 11.4) for the treatment group and 11.7 g/dL (95% CI, 11.3 to 12.0) for the control group (Table 2). After propensity weighting and adjusting for baseline hemoglobin and number of units transfused, the mean difference in hemoglobin at 12 weeks was significantly higher in the IV iron group compared to control group (intergroup difference 0.6 g/dL; 95% CI, 0.1 to 1.1; *p* = 0.01) (Table 2). The MCV at 12 weeks for the patients who received IV iron was 84.5 μm^3^ (95% CI, 82.4 to 86.5), while those who were not treated had an MCV of 85.8 μm^3^ (95% CI, 84.6 to 87.1). The mean difference in MCV at 12 weeks was not significant after adjustment and weighting (intergroup difference 0.7 μm^3^; 95% CI, −1.3 to 2.7; *p* = 0.50).

At baseline, without adjustments, 51% of the treated and 74% of the untreated were in NYHA class I or II (*p* = 0.006) (Figure 2). Twelve weeks after diagnosis, 75% of the treatment group and 70% of the control group were in NYHA class I or II. Adjusting for baseline NYHA class, units transfused, and inverse probability weights, the distribution of NYHA classes among each group was significantly different. The odds ratio for improvement compared to the control group was 2.84 (95% CI, 1.42 to 5.68; *p* = 0.003).

### 3.3. Adverse Events and Safety Outcomes

At 12 weeks, 2 patients in the treatment group and 1 patient in the control group were deceased (Figure 2). Additionally, there were no differences in the adverse events between the groups after adjusting with inverse probability weight of the propensity score. The hazard of hospitalization was similar between the two groups (HR 1.15; 95% CI, 0.55 to 2.42; *p* = 0.71) (Figure 3). Likewise, there was no difference in hazard of hospitalization from any cardiovascular cause (HR 0.55, 95% CI, 0.14 to 2.22; *p* = 0.40) (Table 3). Adverse events were similar, including gastrointestinal bleeding (HR 1.67; 95% CI, 0.56 to 5.37; *p* = 0.39) and infection (HR 0.50; 95% CI, 0.12 to 2.05; *p* = 0.34). Time to event rates of arrhythmia (HR 1.94; 95% CI, 0.59 to 6.40; *p* = 0.28) were comparable among the treated and untreated. Of note, there were no anaphylaxis nor VAD pump thrombosis events documented in either group (Table 3).

## 4. Discussion

This study evaluates the effectiveness and safety of IV iron therapy in end-stage heart failure patients on durable LVAD support. Iron deficiency is common among LVAD patients, with nearly 40% of our cohort demonstrating evidence of iron deficiency. Our main finding is that treatment with IV iron sucrose in LVAD patients with iron deficiency was associated with improvements in hemoglobin and functional status, but not in MCV. Similar adverse event profiles were noted between the two groups.

Iron deficiency is widely present in patients with heart failure, with an estimated prevalence near 50% [7]. It is an independent predictor of worse functional capacity [18]. Similarly, patients supported by durable mechanical support also have elevated rates of iron deficiency, reported to be greater than 60% [15]. Despite limited data describing safety and efficacy of IV iron therapy in advanced heart failure patients on durable VAD support, it has become common practice to treat these patients with IV iron. With the lack of randomized controlled trials to evaluate the benefit of IV iron therapy in the VAD patient population, we aimed to answer the questions of safety and effectiveness outcomes via a retrospective, inverse probability weighted analysis.

After controlling for potential confounders, our study demonstrates a significant increase in functional status. Our results align with previous studies showing improvement in the general heart failure population [11,12]. The Ferinject Assessment in Patients with Iron Deficiency and Chronic Heart Failure (FAIR-HF) trial evaluated the effects of IV ferric carboxymaltose in patients with chronic heart failure with reduced ejection fraction and evidence of iron deficiency [12]. According to the study, patients who were treated with IV ferric carboxymaltose experienced improvement in NYHA functional class. Similarly, the Beneficial Effects of Long-term Intravenous Iron Therapy with Ferric Carboxymaltose in Patients with Symptomatic Heart Failure and Iron Deficiency (CONFIRM-HF) trial also evaluated the effects of IV ferric carboxymaltose in patients with chronic heart failure with reduced ejection fraction [11]. Patients demonstrated improvements in 6-minute walk test, Patient Global Assessment, and NYHA functional class when treated with IV ferric carboxymaltose. While these studies included mostly stage C heart failure patients and our study included stage D heart failure on mechanical support, the results were comparable with significant improvements in NYHA class in both studies. We realize the NYHA class is a subjective estimate of a patient’s functional ability. The NYHA class was chosen in this study as it was available in notes for each patient as it is a part of the general template for the VAD patient population. Unfortunately, more objective measures, including 6-minute walk test and the Kansas City Cardiomyopathy Questionnaire, were not widely available for these patients.

The mean effect of IV iron on hemoglobin was also similar to previous studies. Patients in our cohort treated with IV iron demonstrated a 0.6 g/dL greater improvement in hemoglobin at 12 weeks than those treated not receiving IV iron when adjusted for baseline. Comparing treatment to placebo, FAIR-HF demonstrated a significant mean difference in hemoglobin 1.06 g/dL. Similarly, a significant mean difference in hemoglobin of 0.6 g/dL was demonstrated in CONFIRM-HF at 24 weeks [11]. The significant improvement in hemoglobin may mediate the improvement in NYHA class demonstrated in our study.

Interestingly, our study did not demonstrate a significant mean effect of IV iron on MCV at 12 weeks when adjusted for baseline. FAIR-HF reported a significant difference in MCV of +2.4 μm3 at 12 weeks [12], while MCV was not reported in CONFIRM-HF [11]. It is possible that our study was not powered for this secondary endpoint, but the higher rates of hemolytic anemia in LVAD patient population could also have an effect on the MCV. Given the high rates of gastrointestinal bleed in the mechanical support population, continued slow or intermittent gastrointestinal bleeds may hinder MCV improvement as well.

Separately, there is a small prospective, observational study which compared IV ferric gluconate or ferumoxytol and oral iron among patients supported by LVAD [19]. There was no difference in hemoglobin change (+2.2 vs. +1.8 g/dL; *p* = 0.386). Changes in MCV were not reported. There was also no significant difference in NYHA functional class [19]. The discrepancy may be due to lack of power of the prospective study given small sample size, as only ten patients were in the treatment group. Furthermore, there were differences among baseline variables between the IV and oral iron groups, which may make functional class and hemoglobin changes difficult to compare.

Regarding safety outcomes, patients with and without treatment had similar hazard of hospitalizations, including hospitalizations from any cardiovascular cause. Adverse events among the groups were not significantly different. Although the two groups had similar hospitalization rates at 90 days, it is important to note that they were near 40%. Similarly, the incidence of gastrointestinal bleeds, infections, and arrhythmias were each near 10% at 90 days. Although IV iron appears to improve NYHA class, this confirms the significant burden LVAD patients have on the health system and clear need for novel interventions.

## 5. Limitations

Despite efforts to control for confounding variables, as this study was retrospective, the decision to treat with IV iron could have been impacted by factors that could not adequately be controlled. Additionally, our study may have insufficient power to detect small changes in effectiveness and safety endpoints associated with use of IV iron, which may be manifested in the relatively wide confidence intervals around point estimates for the hazard ratio for gastrointestinal bleeding and arrhythmias. A larger sample size in future studies would reduce the type II error rate. In this study, very few patients had hemoglobin concentrations in the normal range. Thus, we were unable to compare the impact of IV iron therapy in patients with and without anemia, as was conducted by both the FAIR-HF and CONFIRM-HF trials where benefit was observed irrespective of anemic status. Lastly, repeat ferritin and transferrin levels were not routinely performed in practice, so it is unclear if IV iron therapy successfully corrected iron deficiency in this cohort. This confirms the need for randomized controlled trials to assess the effectiveness of IV iron in this vulnerable patient population.

## 6. Conclusions

In patients with LVADs and iron deficiency, treatment with IV iron sucrose was safe and associated with improvements in functional status and hemoglobin.

## Figures and Tables

**Figure 1 jcm-11-03900-f001:**
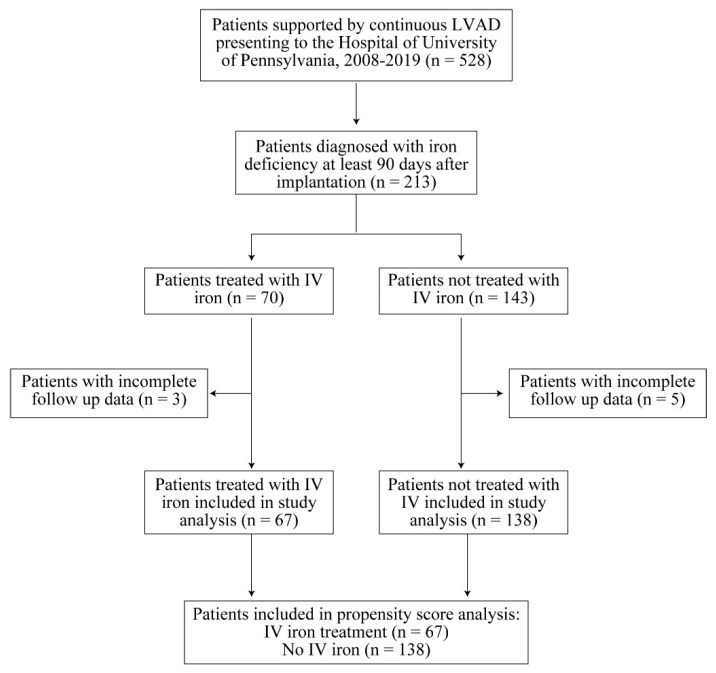
Patient Population.

**Figure 2 jcm-11-03900-f002:**
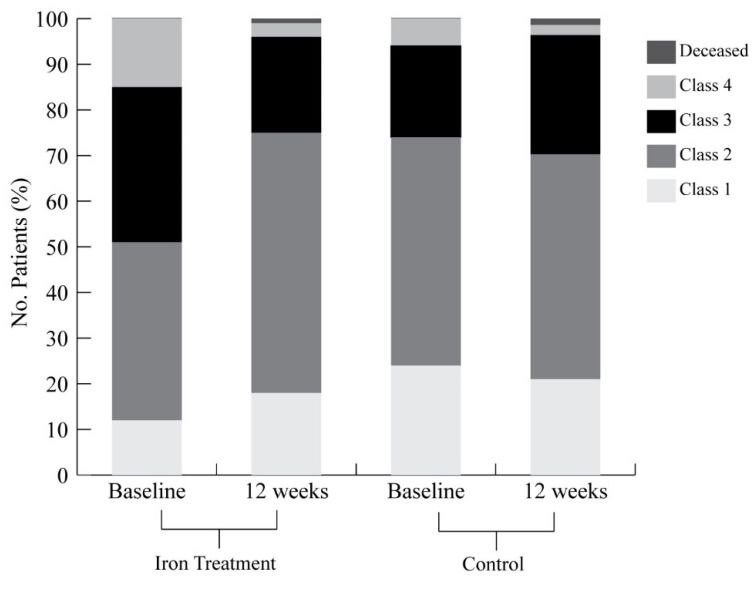
NYHA class at baseline and 12 weeks for the iron treatment group and the control group are demonstrated. The odds ratio of improvement as compared to the control was 2.84 (95% CI, 1.42 to 5.68; *p*-value 0.003). The odds ratio was adjusted for the inverse probability weight of the propensity score, baseline NYHA class, and units of packed red blood cells transfusion between baseline NYHA and week 12. At week 12, 2 patients in the iron treatment group and 1 patient in the control group were decreased.

**Figure 3 jcm-11-03900-f003:**
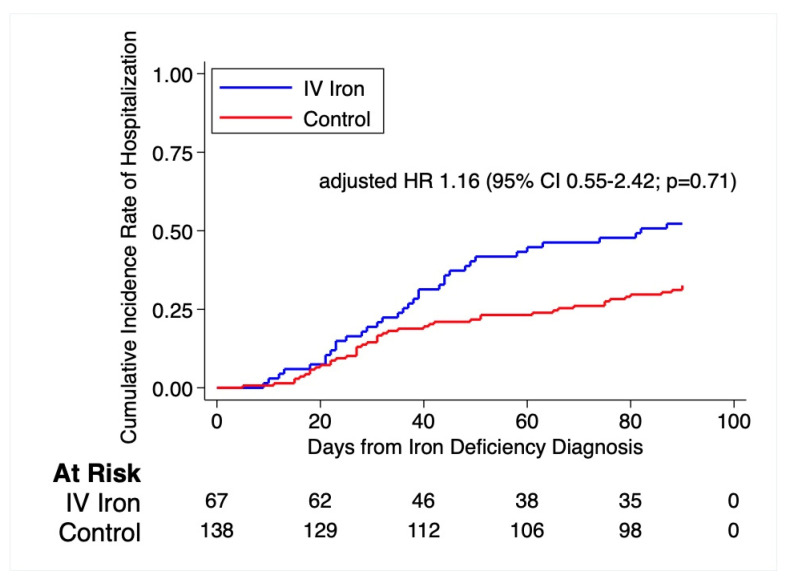
The Kaplan-Meier curve demonstrates the event rate of first hospitalization. Adjusting for the inverse probability weight of the propensity score, the time to first hospitalization hazard ratio as compared to the control group was 1.16 (95% CI 0.55–2.42; *p*-value = 0.71). Log rank *p*-value prior to adjustment = 0.006.

**Table 1 jcm-11-03900-t001:** Baseline Characteristics before Inverse Probability Weighting †.

Characteristic	Iron Treatment (*n* = 67)	Control (*n* = 138)	*p*-Value
**Age—yr**	62.0 (54.0, 69.0)	59.0 (50.0, 68.0)	0.15
**Male sex—No. (%)**	56 (83.6%)	119 (86.2%)	0.61
**Race—No. (%)**			0.10
White	34 (50.7%)	89 (64.5%)	
Black	30 (44.8%)	41 (29.7%)	
Other	3 (4.5%)	8 (5.8%)	
**BMI—kg/m^2^ ‡**	31.7 (26.3, 36.4)	29.8 (25.6, 34.3)	0.23
**NYHA Class—No. (%)**			0.006
I	8 (11.9%)	33 (23.9%)	
II	26 (38.8%)	69 (50.0%)	
III	23 (34.3%)	28 (20.3%)	
IV	10 (14.9%)	8 (5.8%)	
**Inpatient status—No. (%)**	46 (68.7%)	43 (31.2%)	<0.001
**Ischemic etiology—No. (%)**	27 (40.3%)	56 (40.6%)	0.97
**LVAD type—No. (%)**			0.77
Heartmate II	16 (23.9%)	38 (27.5%)	
Heartmate III	37 (55.2%)	69 (50.0%)	
Heartware HVAD	14 (20.9%)	31 (22.5%)	
**Goals of therapy—No. (%)**			0.77
Bridge to transplant	18 (26.9%)	40 (29.0%)	
Destination	48 (71.6%)	94 (68.1%)	
Bridge to recovery	1 (1.5%)	4 (2.9%)	
**Medical history—No. (%)**			
Gastrointestinal bleed	44 (65.7%)	52 (37.7%)	<0.001
Diabetes mellitus	26 (38.8%)	42 (30.4%)	0.23
COPD	7 (10.4%)	13 (9.4%)	0.82
**Laboratory measurements**			
Hemoglobin—g/dL	9.2 (8.5, 10.6)	11.0 (9.6, 12.8)	<0.001
Mean corpuscular volume—μm^3^	79.0 (73.0, 86.0)	86.0 (81.0, 90.0)	<0.001
Serum ferritin—μg/L	46.0 (30.0, 117.0)	71.0 (45.0, 104.0)	0.049
Aspartate aminotransferase—U/L	19.0 (15.0, 29.0)	23.0 (20.0, 29.0)	0.005
Alanine aminotransferase—U/L	15.0 (11.0, 24.0)	19.0 (15.0, 26.0)	0.008
Creatinine—mg/dL	1.3 (1.0, 1.8)	1.2 (1.0, 1.5)	0.044
**Concomitant treatment—No. (%)**			
ACE inhibitor or ARB	33 (49.3%)	103 (74.6%)	<0.001
Beta-blocker	50 (74.6%)	99 (71.7%)	0.66
Digoxin	18 (26.9%)	45 (32.6%)	0.40
Antiplatelet therapy	46 (68.7%)	110 (79.7%)	0.082
Anticoagulant therapy	60 (89.6%)	132 (95.7%)	0.093
Proton pump inhibitor	48 (71.6%)	80 (58.0%)	0.058
H2 blocker	13 (19.4%)	28 (20.3%)	0.88
Oral iron	23 (34.3%)	54 (39.1%)	0.51
**Time between LVAD implant and hemoglobin—days**	575.0 (232.0, 904.0)	377.5 (203.0, 655.0)	0.026

† Parentheses values are medians (interquartile ranges) for continuous variables and number patients (percent) for binary variables. ACE denotes angiotensin-converting enzyme, ARB angiotensin-receptor blocker, NYHA New York Heart Association, and COPD chronic obstructive pulmonary disease. ‡ The body mass index is the weight in kilogram divided by the square of height in meters. History of diabetes mellitus and COPD were defined by problem list and current medication. History gastrointestinal bleed was determined by history of gastrointestinal bleed listed as problem list or admission.

**Table 2 jcm-11-03900-t002:** Primary and Secondary End Points at 12 weeks.

Variable	Iron Treatment Mean (95% CI)	Control Mean (95% CI)	Mean Difference or Odds Ratio Adjusted for Baseline	*p*-Value †
Hemoglobin—g/dL	11.0 (10.5–11.4)	11.7 (11.3–12.0)	0.6 (0.1–1.1) *	0.01
MCV—μm^3^	84.5 (82.4–86.5)	85.8 (84.6–87.1)	0.7 (−1.3–2.7) *	0.50
NYHA Class—%			2.84 (1.42–5.68) º	0.003
I	17.9	21.0		
II	56.7	48.3		
III	20.9	26.1		
IV	3.0	2.2		

* Denotes mean difference (95% confidence interval) adjusted for corresponding baseline variable and inverse probability weight of propensity score. º Denotes odds ratio for improvement (95% confidence interval) adjusted for corresponding baseline variable and inverse probability weight of propensity score. † *p*-value is for the corresponding adjusted mean difference or odds ratio.

**Table 3 jcm-11-03900-t003:** Hazards of Safety End Point and Adverse Events.

	Iron Treatment	Control		
**Safety end Point**	No. of Patients with End Point or Event (Percent)	No. of Patients with End Point or Event (Percent)	Time to First Event Hazard Ratio (95% CI) *	*p*-value
Hospitalization within 90 days	35 (52.2)	45 (32.6)	1.15 (0.55–2.42)	0.71
Hospitalization for any cardiovascular cause within 90 days	8 (17.9)	11 (8.0)	1.67 (0.52–5.37)	0.39
**Adverse event**				
Gastrointestinal bleed	7 (10.4)	7 (5.1)	1.86 (0.56–6.22)	0.31
Infection	8 (11.9)	13 (9.4)	0.50 (0.12–2.05)	0.34
Arrhythmia	8 (11.9)	6 (4.3)	1.94 (0.59–6.40)	0.28
Pump thrombosis	0 (0.0)	0 (0.0)	1.00	1.00
Anaphylaxis	0 (0.0)	0 (0.0)	1.00	1.00

* Time to first event hazard ratio adjusted for inverse probability weight of the propensity score. *p*-value is for time to first event hazards ratio.

## Data Availability

The data that support the findings are available on request from the first author, C.J.P.

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
