# Peer review of "Safety and Effectiveness of Intravenous Iron Therapy in Patients Supported by Durable Left Ventricular Assist Devices"

_jcm, 2022, doi:10.3390/jcm11133900_

Round 1
Reviewer 1 Report
The authors present an interesting study on the use of IV iron in LVAD patients utilizing a single center experience through EMR review. This study examines an important and unanswered question. The end points examined are also pertinent, with objective (Hgb, MCV) and subjective (NYHA class) components. The authors show that over 12 weeks of follow up, patients receiving IV iron sucrose had a significantly higher Hgb and improved NYHA class compared to matched controls, but no difference in MCV compared to controls.
While this study is a strong effort to provide some data in a "data-free" zone, I am not sure that the methods of this paper are strong enough to answer the questions at hand. The IV iron group had more GI bleed patients included, and it likely that these patients may have received pRBC repletion during the study period as well. While the authors control for use of RBC in the groups, I am not sure this is enough to obviate the effect of RBC transfusion in these patients. I would suggest a sensitivity analysis restricted to those who did not receive RBC repletion during the study period.
Furthermore, I am also concerned about using NYHA in this kind of analysis. The authors need to describe in greater detail how NYHA class was determined for each patient, and given the subjective nature of this end point, I hesitate to make much of the improvement seen in this short period of time. Of note, significantly more patients in the control group were on RAASi, and while they control for this in their analyses, may explain some of the effect.
Minor comments:
Line 92: correct spelling of waved to waived
Line 96: update sentence to state "at 12 weeks following IV iron therapy"
Line 97: I would want more details about how NYHA class was determined. Was this based on provider notes, and did all notes outline NYHA class in these LVAD patients?
Author Response
We would like to thank the editor and the reviewers for reviewing our paper and for the opportunity to revise it. We were able to address most of the reviewers concerns, as detailed below.
Reviewer 1:
- While this study is a strong effort to provide some data in a "data-free" zone, I am not sure that the methods of this paper are strong enough to answer the questions at hand. The IV iron group had more GI bleed patients included, and it likely that these patients may have received pRBC repletion during the study period as well. While the authors control for use of RBC in the groups, I am not sure this is enough to obviate the effect of RBC transfusion in these patients. I would suggest a sensitivity analysis restricted to those who did not receive RBC repletion during the study period.
Thank you for this important comment. We had numerous discussions about this point, and we understand the concern. However, in the statistical analysis we controlled for pRBC, which we believe is sufficient. - Furthermore, I am also concerned about using NYHA in this kind of analysis. The authors need to describe in greater detail how NYHA class was determined for each patient, and given the subjective nature of this end point, I hesitate to make much of the improvement seen in this short period of time. Of note, significantly more patients in the control group were on RAASi, and while they control for this in their analyses, may explain some of the effect.
We explained where we received the NYHA class from – it is a part of the template, completed by our advanced heart failure attending physicians. Unfortunately, more objective measures such as the 6MWT and KCQS were not readily available for patients.
We realize the groups were different at baseline, but the weight propensity scores should mitigate the baseline variables affect on outcomes. After weighting, the differences between each variable were <0.05.
Minor comments (completed all of these)
Line 92: correct spelling of waved to waived
Line 96: update sentence to state "at 12 weeks following IV iron therapy"
Line 97: I would want more details about how NYHA class was determined. Was this based on provider notes, and did all notes outline NYHA class in these LVAD patients?
Sincerely,
Edo Y. Birati, MD FACC
Associate Professor of Medicine
Reviewer 2 Report
In the present study, Drs Peters, Birati and colleagues, evaluated the safety and effectiveness of IV iron therapy in patients supported by LVADs with iron deficiency.
The manuscript is clear, well written and interesting.
A few minor issues need to be addressed:
1. In the methods section second paragraph (page 2 line 79): Although difficult to define in a retrospective study, the authors are encouraged to describe, if known, the reasons for not treating LVAD supported patients having IDA with IV Iron and if unknown, elaborate on the potential causes for this discrepancy between the two groups.
2. From Table 1, despite "All of baseline characteristics listed in Table 1 were used in the propensity score and, after weighting, resulted in the cohorts having no significant differences between them with p-values >0.05.", the treated group is indeed a sicker cohort, on longer support and with more comorbidities such as anemia (microcytic..) GI bleeds, Hypertension (or other causes for ACEi treatment) and twice as more in NYHA III-I than the untreated group..
3. The comparison between the two groups should not be the main endpoint of the study but rather the safety and efficacy of IV iron Tx for LVAD supported patients with IDA.
4. Safely improvements in functional class and Hb levels in the treated group is by itself an interesting finding suggesting that by supplying IV iron to the LVAD patient with iron deficiency the patient's functional class levels with the LVAD supported patient that has no iron deficiency.
5. The occurrence of CVA in both groups: hemorrhagic or thrombotic was not addressed.
Author Response
We would like to thank the editor and the reviewers for reviewing our paper and for the opportunity to revise it. We were able to address most of the reviewers concerns, as detailed below.
Reviewer 2:
- In the methods section second paragraph (page 2 line 79): Although difficult to define in a retrospective study, the authors are encouraged to describe, if known, the reasons for not treating LVAD supported patients having IDA with IV Iron and if unknown, elaborate on the potential causes for this discrepancy between the two groups.
We thank the reviewer for this important comment. The reasons certain patients were not treated are unknown, but likely patients who had more significant iron deficient anemia (lower hemoglobins, lower MCVs, lower ferritin) were chosen to get IV iron. Also patients with history of GI bleeds were likely to get IV iron. Likely because these more significant lab values and co-morbidities may have triggered the physician to order the medication. We added this to our manuscript.
- From Table 1, despite "All of baseline characteristics listed in Table 1 were used in the propensity score and, after weighting, resulted in the cohorts having no significant differences between them with p-values >0.05.", the treated group is indeed a sicker cohort, on longer support and with more comorbidities such as anemia (microcytic..) GI bleeds, Hypertension (or other causes for ACEi treatment) and twice as more in NYHA III-I than the untreated group..
We understand there are significant differences in the baseline characteristics of the two cohorts. To account for these systematic differences, we performed a propensity score with weighting to make the two cohorts balanced and comparable. After weighting, the two cohorts are similar with p values all less than 0.05. It is typical to present the baseline characteristics as such before matching, but let us know if there is another way you’d like the data presented.
- The comparison between the two groups should not be the main endpoint of the study but rather the safety and efficacy of IV iron Tx for LVAD supported patients with IDA.
We decided to compare treated vs not treated because many of these patients were hospitalized or otherwise may have been ill at the time, and we felt that patients’ hemoglobin, NYHA class, etc may improve with time even if they weren’t treated. That is the case as we report that patients without IV iron also had some improvement in their endpoints. We still report the changes in the endpoints for patients with IV iron individually, but without comparing it to patients without IV iron, we felt that readers would want to know if there was difference when compared no treatment. We also mirrored our endpoints to the FAIR-HF trial which reported intergroup differences similar to ours.
- Safely improvements in functional class and Hb levels in the treated group is by itself an interesting finding suggesting that by supplying IV iron to the LVAD patient with iron deficiency the patient's functional class levels with the LVAD supported patient that has no iron deficiency.
I am so sorry, but I am not sure if I follow this comment. We don’t present data within this cohort of patients without iron deficiency.
- The occurrence of CVA in both groups: hemorrhagic or thrombotic was not addressed.
We did not include occurrence of CVA as an adverse event as it was a small follow up study as this hasn’t been a known postmarketing adverse reaction of IV iron (unlike hypersensitivity, infection, arrhythmias, GI upset, etc). A sentence explaining this was included. If you feel strongly to include CVA as an adverse reaction, please let us know. I did not include it in this version as it would require a significant amount of chart review.
Sincerely,
Edo Y. Birati, MD FACC
Associate Professor of Medicine